# Temperature Uncertainty Reduction Algorithm Based on Temperature Distribution Prior for Optical Sensors in Oil Tank Ground Settlement Monitoring

**DOI:** 10.3390/s24072341

**Published:** 2024-04-07

**Authors:** Tao Liu, Tao Jiang, Gang Liu, Changsen Sun

**Affiliations:** College of Optoelectronic Engineering and Instrumentation Science, Dalian University of Technology, Dalian 116024, China; liutao2020@mail.dlut.edu.cn (T.L.); jiangtao19980125@mail.dlut.edu.cn (T.J.); uiliugang@mail.dlut.edu.cn (G.L.)

**Keywords:** optical fiber sensor, ground settlement monitoring, temperature uncertainty, temperature distribution, numerical simulation

## Abstract

Ground settlement (GS) in an oil tank determines its structural integrity and commercial service. However, GS monitoring faces challenges, particularly due to the significant temperature differences induced by solar radiation around the tank in daytime. To address this problem, this paper digs out a prior and proposes a temperature uncertainty reduction algorithm based on that. This prior has a spatial Gaussian distribution of temperature around the tank, and numerical simulation and practical tests are conducted to demonstrate it. In addition, combining uniformly packaged sensor probes and the spatial prior of temperature, the temperature uncertainty is verified to be Gaussian-distributed too. Then, the overall temperature uncertainty can be captured by Gaussian fitting and then removed. The practical test verified a 91% reduction rate in temperature uncertainty, and this approach enables GS sensors to effectively perform daytime monitoring by mitigating temperature-related uncertainties.

## 1. Introduction

GS in oil tanks can lead to serious safety hazards such as structural damage during service and production processes [1,2,3,4]. This can cause the leakage of liquid oil and oil vapor. The leakage of oil can pollute the soil and groundwater [5] and the leakage of oil vapor can pollute the air and cause cancer to develop in workers [6]. To avoid these risks, effective GS monitoring is a routine necessary task in service and many methods have been proposed [7,8,9]. There are mainly two approaches in GS monitoring research. One is the remote sensing approach that utilizes equipment such as Total Stations to monitor the height changes among test points. This method offers the advantage of flexibility and safety, as it does not require contact measurements. However, it is vulnerable to environmental interferences like rain, snow, and low light conditions at night [10,11]. In order to overcome these limitations, researchers have utilized the hydrostatic leveling system (HLS) technique. But, in the oil tank application, two main challenges need to be addressed for the HLS: ensuring fireproof safety and avoiding temperature interference [12].

Facing these two challenges, our research group firstly introduced low-coherence interferometry into the field of oil tank GS monitoring [13]. This approach effectively addressed the first problem by separating the electrical components within the control center from the non-electrical elements which are installed around the tank. However, the temperature uncertainty, especially the temperature uncertainty arising from the HLS scheme, is still unresolved.

In this study, we initially focused on analyzing the temperature distribution surrounding the oil tank. Previously, this algorithm was studied in the time domain by building a mapping relationship between temperature and temperature uncertainty [13]. However, this approach proved to be time-consuming and resource-intensive. Through careful observation and analysis, we discovered that the primary factor contributing to temperature differences was the uneven distribution of sunlight. To gain a deeper understanding, we conducted extensive research on solar radiation and performed simulations to establish a temperature distribution model. Our findings revealed that the temperature around the tank followed a Gaussian distribution pattern with respect to the angle, which was further validated by comparing it with actual temperature data. Additionally, we ensured the uniformity of the sensor probes by selecting highly identical sensors that exhibited consistent scale and shape. Before installation, we utilized Total Station measurements to ensure that all sensors were positioned at the same initial height. This approach created a new prior condition in which all sensor probes exhibited identical temperature responses to the same temperature interference.

Building upon the prior condition of the Gaussian distribution of temperature, we proposed a temperature uncertainty reduction algorithm. To validate its effectiveness, we conducted a practical test. The results show that the algorithm successfully reduced the temperature uncertainty by 91% which effectively mitigated the effects of the differential temperature field.

## 2. Theoretical Background

In this section, a brief introduction to the GS sensor is provided. Then, the temperature uncertainty and its relationship with the unbalanced temperature field are discussed in detail, incorporating the topic of solar radiation.

### 2.1. The Configuration of the Optical GS Sensor

The key point of GS measurement is finding a stable reference point. For Total Stations, a remote benchmark can be easily used as a reference. In the case of HLS-based sensors, the common liquid level serves as a reference point for comparing the position changes of each test point. By continuously measuring the liquid level change from the top of the sensor probe to the liquid level, the relative GS of each test point can be determined compared with the reference point. The relative settlement can be calculated from the following equation [12,14]:(1)$$H_{i}-H_{ref}=\left(K_{i}^{\prime }-K_{i}\right)-\left(K_{ref}^{\prime }-K_{ref}\right)$$
where $H_{i}$ is the GS of the oil tank measured by the installed sensor and indexed by i, corresponding to the sensor location; $H_{ref}$ is the GS of the reference sensor; $K_{i}$ is the distance from the top of the sensor probe to the initial liquid level inside sensor i; and $K_{i}^{\prime }$ represents the distance from top of the sensor probe to the final liquid level inside sensor i after the GS occurs. $K_{ref}$ and $K\prime_{ref}$ are counterparts of reference sensor, respectively. The reference sensor was fixed on a stable benchmark, and in practice, $H_{ref}$ was considered to be zero. By collecting the measured data from all sensors, the overall GS of the oil tank can be evaluated. So, the GS measurement can be mapped to a liquid level measurement problem.

An optical schematic of the interferometer is shown in Figure 1. Briefly speaking, it consists of a reference arm with a variable optical path and a measuring arm with a constant optical path. The motor of the reference arm is driven to scan. When AB and CD are of equal length, a burst peak of the influence fringes can be detected, which serves as the readout for the liquid surface location D and is then related to the GS information.

The theoretical derivation can be found in Ref. [15], and for convenience, it is copied here:(2)$$I\left(Q\right)=I^{\left(1\right)}\left(Q\right)+I^{\left(2\right)}\left(Q\right)+2\sqrt{I^{\left(1\right)}\left(Q\right)}\sqrt{I^{\left(2\right)}\left(Q\right)}\gamma_{12}^{r}\left(\frac{L_{AB}-L_{CD}}{c}\right)$$
where $\gamma_{12}^{r}\left(\frac{L_{AB}-L_{CD}}{c}\right)$ denotes the real part of the complex degree of coherence $\gamma_{12}$, which is a function of the optical path difference; $L_{AB}$ represents the optical path difference determined by the half-reflection mirror A with the moving reflector B; and $L_{CD}$ is from the half-reflection mirror C and the location of the liquid surface D. When $L_{AB}=L_{CD}$, $\gamma_{12}^{r}\left(\frac{L_{AB}-L_{CD}}{c}\right)=\gamma_{12}^{r}\left(0\right)$, which indicates the detection of a peak of the influence fringes; c is the velocity of light; $I^{\left(1\right)}\left(Q\right)$ and $I^{\left(2\right)}\left(Q\right)$ represent the optical intensities from the reference arm and sensing each output branch of the switcher, for example, respectively.

The designed GS sensor can remove environmental interferences from the temperature or other vibrations by introducing the half-reflection mirror in both the reference arm and each output branch of the switcher [12]. This increases the system stability during the liquid level interrogation process. However, the HLS, which consists of GS sensors and connecting tubes, can introduce fluctuations for liquid thermal expansion.

### 2.2. Source of Temperature Uncertainty

The HLS is based on the equivalent liquid pressure at the bottom of the connected tanks, and the Bernoulli equation law can be used to describe the fluid [9] in the system as:(3)$$\frac{1}{2}\rho v^{2}+P_{air}+\rho gH=const$$
where ρ is the density of liquid, v is the velocity of liquid flow, $P_{air}$ is the outer atmospheric pressure, g is the gravitational acceleration, and H is the liquid level from the surface to the bottom of the container. This equation can be simplified by assuming that the liquid in the tube is stationary and the atmospheric pressure is equal in the workspace when considering the quasi-static state of the HLS.

To analyze the temperature interference on the HLS, a simple model is established with two interconnected containers through a liquid tube, as shown in Figure 2a. Initially, both tanks are in the same temperature environment with equal liquid heights. The left and right tanks are labeled as tank L and tank R, respectively, and the common liquid level is denoted as H. When the L tank is heated to a higher temperature, a transient analysis of the fluid model can be used. For easy calculation, let us assume that the two tanks are initially isolated. In this case, the liquid pressure at the bottom of the R tank, represented as $P_{R}$, can be calculated as $P_{R}=\rho gH$, where ρ is the density of the liquid and g is the gravitational acceleration.

After the temperature is increased in the L tank from $T_{0}$ to $T_{1}$, the pressure at the bottom of the L tank can be expressed as $P_{L}^{\prime }=\rho^{\prime }gH^{\prime }$, in which the density, $\rho^{\prime }$, and the liquid level, $H^{\prime }$, can be impacted by the temperature change. To investigate the pressure variation, a more essential formula is needed, $P=\rho gh=\frac{\rho shg}{s}=\frac{\rho Vg}{s}=\frac{mg}{s}$, which means a gravitational force, mg, being applied on a cross-sectional area, s. The parameters in the equation above are represented as follows: h is the liquid level, V is the volume of liquid, and m is the weight of liquid. If the weight m remains constant, the pressure P at the bottom of the liquid will remain unchanged. Consequently, the liquid in the HLS remains stationary, and only the liquid level changes with the temperature fluctuations.

In practice, as shown in Figure 2b, eight sensors(#1–#8) are installed evenly around the oil tank and one reference sensor(#9) is installed far away from the tank acting as a reference. The standard 100,000 $m^{3}$ oil tank has a diameter of 80 m and a height of 21.8 m. This large spatial dimension makes the oil tank suffer from unbalanced sunshine and leads to a nonuniform temperature field, as shown in the picture view of Figure 2c. When the sunlight is blocked by oil tank, the sunlit front can be much hotter than the shadow zone. Considering the installation positions of the sensors, they can be affected by these temperature differences. This is how the temperature uncertainty is introduced. Before analyzing and simulating the temperature distribution, solar radiation must be investigated.

### 2.3. Solar Radiation Model

There are three modes of heat exchange between the oil tank and the surrounding air under solar radiation [16]. In this section, the geometry of solar radiation is introduced in detail for further numerical simulation.

All objects emit and absorb energy in the form of radiation, and the process is directly determined by the relative temperature difference. The surface temperature of the sun is about 5760 K, emitting substantial radiation energy outward. Part of the radiation reaches the Earth’s surface directly and the other part is absorbed, reflected, and refracted by the atmosphere. The radiation that directly reaches the Earth’s surface without being absorbed or scattered is called direct radiation ($G_{ND}$). The radiation that undergoes refraction, scattering, and re-emission is called scattered radiation ($G_{d}$: diffuse radiation). The radiation reflected from one surface to another in close proximity is called reflected radiation ($G_{R}$). So, the total amount of solar radiation received by objects on the Earth’s surface is [17]
(4)$$G_{t}=G_{ND}+G_{d}+G_{R}$$

For the long distance between the sun and Earth, the amount of solar radiation per unit area projected onto the upper boundary of the atmosphere can be seen as constant, independent of the position on Earth and time. For evaluation of the thermal impact on structures and buildings, the solar constant can be seen as a constant as follows [18]:(5)$$I_{sc}=1367W/m^{2}$$

The intensity of solar radiation received by ground objects depends on the geometric position of the sun with respect to the ground, the terrain in which the object is located, the number of clouds in the sky, and other nonuniform atmospheric factors. Here is a brief introduction to geometric quantities related to solar radiation introduced as Figure 3 including various angles related to solar radiation [19,20].

In Figure 3a above, the geographic latitude l of point P is the angle between OP and its projection onto the equatorial plane. The declination angle δ is the angle between the line connecting the center of the sun and the center of the Earth and its projection onto the equatorial plane. The hour angle h is the angle between two lines on the equatorial plane indicating the change in position of the sun in one day. One of the lines is the projection of point P on the equatorial plane and the other is the projection of the line connecting the center of the sun and the center of the Earth onto the equatorial plane.

In Figure 3b, the solar altitude angle β means the projection angle of sunlight and the horizontal plane. According to the geometric relationships of incident sunlight, it can be inferred that
(6)sinβ=coslcoshcosδ+sinlsinδ

Angle β can reach the maximum at high noon:(7)$$\beta =90^{o}-\left|l-\delta \right|$$

Another angle known as the solar azimuth angle ϕ is the clockwise angle between the projection of sunlight onto the horizontal plane and the north direction. From the geometrical relationship, the angle ϕ locating the solar position can be expressed by introducing the three locating angles:(8)$$cos\phi =\frac{sin\delta cosl-cos\delta sinlcosh}{cos\beta }$$

This formula represents how light energy is projected onto planes. When dealing with the complex curvature plane, the projection relationship can be altered with light projected in the normal line direction. To enlarge the application of any surface, an angle factor called the surface solar azimuth angle γ is introduced, which represents the angle between the line of sunlight projected onto the horizontal surface and the normal line of a surface projected onto the same horizontal surface, mainly used to locate the position of an arbitrary surface. Also, the surface azimuth angle ψ is the angle between the projection of a normal line onto the horizontal plane and the north direction clockwise. Apparently, we can infer that as follows:(9)$$\gamma =\left|\phi -\psi \right|$$

When calculating the projection of sunlight onto the surface, the angle of incidence θ is the angle between sunlight and the normal line of the surface. The tilt angle α is the angle between the normal line of the horizontal plane and the normal line of the surface. So, when calculating the projection, the corresponding projection angle can be expressed as follows:(10)cosθ=cosβcosγsinα+sinβcosα

Especially, when the surface is a vertical plane:(11)cosθ=cosβcosγ

When the surface is a horizontal plane:(12)cosθ=sinβ

After substituting Equation (9) into Equation (10), the angle factor only consists of ϕ, ψ, β, and α, which are the required angle variables to be used when calculating the solar radiation.

In practice, when considering solar radiation and the induced temperature field, some boundary conditions have to be considered. The oil tank is a solid structure primarily made of steel, and the overall heat flow on its surface is the sum of the total heat flow, $q_{tot}$, which is the sum of heat convection ($q_{c})$, heat radiation ($q_{rad}$), and solar radiation ($q_{solar}$) [17]:

### 2.4. Heat Transfer Model


(13)
$$q_{tot}=q_{c}+q_{rad}+q_{solar}$$


The rate of heat flow due to convection, $q_{c}$, can be calculated as follows [21]:(14)$$q_{c}=hA\left(T_{c}-T_{a}\right)$$
where $T_{c}$ represents the surface temperature of the oil tank, A is the surface area of the tank, $q_{c}$ is the heat transferred per unit time, and h is a coefficient that depends on the material properties of the oil tank, the surrounding air, and factors such as air flow speed.

Thermal radiation does not require a medium for energy transfer. The rate of heat radiation, $q_{rad}$, can be mathematically expressed as $q_{rad}=\varepsilon \sigma A\left(T_{c}^{4}-T_{a}^{4}\right)$. In this formation, ε is the emissivity of the material, A is the surface area, $T_{c}$ and $T_{a}$ are the temperatures of the emitting and receiving surfaces, and σ is the Stefan–Boltzmann constant ($5.670\times 10^{-8}\;W/\left(m^{2}{\;K}^{4}\right)$).

Solar radiation is the primary source of thermal exchange and temperature change. Based on the previous discussion on solar light projection and its variation with location and surface direction, researchers have developed detailed models to describe the process from an intensity perspective. Solar angle information can be obtained from weather station websites, such as the United States Naval Observatory (USNO) website [17]. The intensity of solar radiation, $I_{d},$ is calculated using the empirical model developed. It is described as $I_{d}=I_{0}\;e^{-cm}$, where $m=\frac{1}{sin\left(l\right)}$ represents the optical air mass indicating the atmospheric condition of the sunlight path, c is the atmospheric extinction coefficient, and $I_{0}$ is the extraterrestrial solar illuminance given by $I_{0}=I_{sc}\left(1+0.034cos\left(\frac{2\pi }{365\left(n-2\right)}\right)\right)$, where $I_{sc}$ is the solar constant mentioned earlier and n represents the day of the year ranging from 1 to 365 [22].

Considering the geometry, the solar radiation normal to a panel can be expressed as $I_{n}=I_{d}\;cos\theta$. After calculating all the energy from solar radiation, we need to consider the solar absorptivity, $A_{ab}$, of the object, which can be influenced by the material and color of the surface coating. The final rate of heat flow caused by solar radiation absorbed by the object’s surface can be determined as $q_{solar}=A_{ab}\;I_{n}\;S$, where S is the surface area of the object.

These three components—convection, radiation, and solar radiation—contribute to the heat transfer and resulting temperature change. The final temperature change is determined by considering the heat capacity of the material and the surrounding atmosphere. With the advance of Finite Element Analysis (FEA), these calculations can be solved with the help of software.

## 3. Algorithm Research

### 3.1. Finite Element Analysis on Temperature Field of Oil Tank

In the field of FEA, there are many pieces of mature software, and, in this study, heat transfer analysis was conducted by using Ansys Fluent [23]. The geometry modeling and finite element mesh generation result are provided in Figure 4. The ordinary oil tank is basically a cylindrical shape and its volume is 100,000 $m^{3}$ with a diameter of 80 m and height of 21.8 m. The Fluent meshing is developed as shown in Figure 4b.

Heat transfer analysis was performed with polyhedral network elements. The temperature gradients were mainly spread along the surface in the height and circumferential directions of the oil tank for the thin shell structure. For a more accurate simulation, three layers were added to grasp the temperature change at a location on the perimeter wall and the top layers connecting with the surrounding air. The detailed parameters of thermal analysis are presented in Table 1. For the changeable surrounding atmosphere and the white color coating, the ratio of acceptance of solar radiation or atmospheric temperature factors needs to be tested using an in-field test.

### 3.2. Result of Simulation

As mentioned above, solar radiation changes every day with geographic latitude l, but the Fluent software provides automatic calculation tools, which can change the inner coefficient automatically. All we need to do is add both spatial and temporal information, such as Dalian city at about $39^{o}\;N,\;120^{o}\;E$ and the day of 22 June.

The result of the temperature distribution on the cylindrical wall is illustrated in Figure 5a at the time of 14:00. The Z coordinate represents the south direction. It can be observed that the southwest direction of the tank receives more energy, resulting in a higher temperature compared to other directions on the vertical surface of the oil tank. This observation is consistent with intuition and qualitative analysis. To further investigate this phenomenon, temperature data from various time points were simulated and are presented in Figure 5b for 8:00, 10:00, 12:00, 14:00, 16:00, and 18:00 on June 22nd.

In order to validate the accuracy of the simulation, the measured temperature data recorded from the distributed Raman temperature measurement system are compared with the simulation results in Figure 5c. The measured temperature data are obtained from eight sensor probes positioned around the oil tank, with a horizontal coordinate interval of 45°. It can be observed that the two lines are highly consistent with each other. In the selected morning, noon, and afternoon time, it is in accordance with the tested temperature information. The largest temperature difference is within the range of 3 °C, but the curve trends are consistent in all the figures. Based on this comparison, we can conclude that the simulation results are quite accurate.

### 3.3. Temperature Uncertainty Reduction Algorithm

Since the Gaussian temperature distribution and the centroid of this distribution, have been revealed, we need to consider how to implement the temperature information to remove the temperature uncertainty. Compared with the test temperature, by using the simulated temperature distribution information, not only can we save money on the temperature recording equipment, but we can also provide more precise centroid information compared with the isolated temperature measurement installed on the sensors.

As for the temperature uncertainty, all the sensor probes are of unified size and shape, and the liquid level is the same, which was guaranteed by the Total Station technique before installation. So, the liquid level change response, which is seen as temperature uncertainty in this paper, is identical. Figure 3 of Ref. [15] shows that the four sensors located around the oil tank in different directions exhibit similar temperature responses with a similar slope, marked as K. Figure 6 demonstrates how the temperature uncertainty varies with the temperature. During nighttime, when the temperature difference at all locations around the tank is zero, the temperature uncertainty is minor. As illustrated in Figure 2a, the increased temperature can induce a rise in liquid level without liquid flowing in or out from the bottom of the liquid tube. That means that in a certain temperature range, we can use a coefficient to describe the change in liquid level with the change in temperature, which is the K value above. That means that if the temperature ($T_{1},\;T_{2}\ldots T_{6}$) around the tank is in a Gaussian distribution, as shown in Figure 6, the temperature uncertainty can also be in the form of a Gaussian distribution by multiplying a common K value in the spatial domain.

Since the prior condition of the temperature uncertainty distribution is in accordance with the temperature difference as the Gaussian distribution, we can plot a Gaussian fitting line and subtract the fitted data from the raw data. Besides the Gaussian distribution, the centroid of symmetry can be known, which can help to reduce the complexity of Gaussian fitting.

The Gaussian curve is a common curve model used to fit data with Gaussian distribution characteristics. In practice, we apply this form to describe the temperature field:(15)$$f\left(x\right)=Je^{-\frac{{\left(x-u\right)}^{2}}{2w^{2}}}$$
where J is the amplitude parameter that controls the peak of the curve, u is the mean parameter that controls the center position of the curve, and w is the standard deviation parameter that controls the width of the curve.

The algorithm starts from the Gaussian distribution function and the centroid of the Gaussian function. Next, initial values are set for the fitting parameters J and w. These initial values serve as starting points for the iterative computation. The algorithm then proceeds with iterative computation, where the values of the fitting parameters J and w are continuously adjusted in order to minimize the sum of squared residuals. This process involves repeatedly updating the parameter values until the minimum sum of squared residuals is reached.

Once the sum of squared residuals reaches its minimum, the optimal fitting parameters can be obtained. With these parameters, the algorithm constructs a best fitting curve, which represents the best fit to the measured raw data.

At the end, the algorithm removes the fitted Gaussian line from the measured raw data, resulting in the final results. This step ensures that the fitted curve is separated from the original data, allowing for a clear analysis of the remaining information. By following these steps above, the algorithm can effectively fit the Gaussian function to the measured temperature data and extract the temperature uncertainty from the measured results.

To verify the performance of the algorithm, an actual experiment is carried out. In Figure 7, the measured GS data in the daytime (14:00 h), the data from the night (24:00 h), and the processed data by the algorithm are plotted together for comparison. The result is compared with the GS data collected at night when there is no spatial temperature difference and can be considered as the ground truth of GS. In Table 2, the GS in the daytime contains temperature uncertainty and GS. And the difference between daytime and night can be seen as temperature uncertainty. The fourth column is the processed data after subtracting the Gaussian fitting line (temperature uncertainty) and can be seen as the ground truth of GS. The last column in the table represents the error of the algorithm. This is calculated as the difference between the ground truth measurement of GS and the calculated GS value.

It can be seen that the curve tends to become much flatter after the process, which means that the influence decreases sharply. It can be seen that by introducing this algorithm, the temperature uncertainty is suppressed sharply. According to the data listed in Table 2, it can be seen that the raw data of the measured GS in the daytime can reach 34.8 mm, and the maximum of the processed data by the algorithm is 3.23 mm, which means that the temperature uncertainty is expressed as a ratio of 91%, which proves the effectiveness of algorithm.

## 4. Conclusions

In this paper, aiming to eliminate the temperature uncertainty in the GS monitoring sensor around a large-scale oil tank, the temperature influence on the HLS was thoroughly investigated from both theoretical and numerical simulations. The simulation results demonstrated the temperature field distribution pattern and temperature difference around the oil tank after fully considering the geometrical factors of solar radiation and the different types of outside factors including materials. After understanding the origin and distribution pattern of the temperature field of solar radiation, the Gaussian distribution and centroid of symmetry of the uncertainty were utilized as the prior condition for the uncertainty reduction algorithm. Practical tests and data analysis were conducted to evaluate its performance. The statistical data denote that our algorithm can suppress the error caused by a nonuniform temperature field by more than 91%. It is proved that the combination of the optical GS sensor with the designed algorithm can effectively reduce the temperature effect present in daytime in-field oil tank GS monitoring.

## Figures and Tables

**Figure 1 sensors-24-02341-f001:**
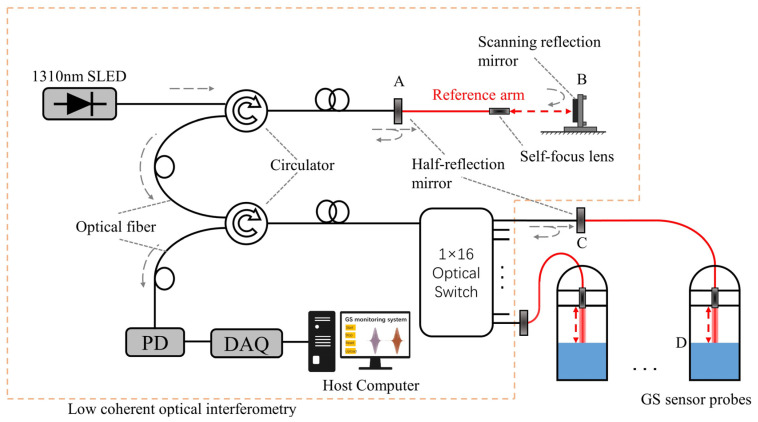
The schematic of the GS sensor configured in a low-coherence fiber-optic Michelson interferometer. SLED, super-luminescent emitting diode; PD, photodetector. The fiber in the experiment is a single-mode fiber (SMF − 28e+) from the Corning company; A and C represent half-reflection mirrors; D indicates the location of the liquid surface, acting as a total-reflection mirror; B is a total-reflection mirror, which is controlled by a stepping motor with an accuracy of 1.25 µm/step. The distance between C and D varies corresponding to GS. The optical path is as illustrated as the arrows.

**Figure 2 sensors-24-02341-f002:**
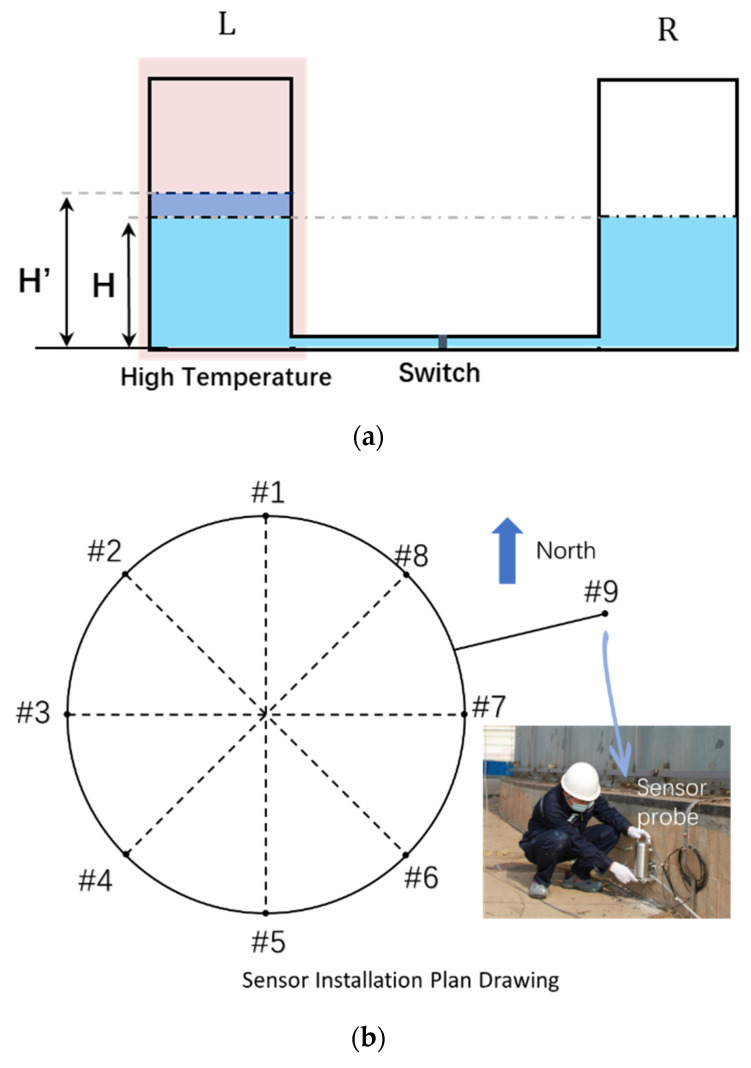
Model of HLS with a nonuniform temperature field and actual installation plot. (**a**) Basic model of HLS; (**b**) sensor installation around the oil tank.

**Figure 3 sensors-24-02341-f003:**
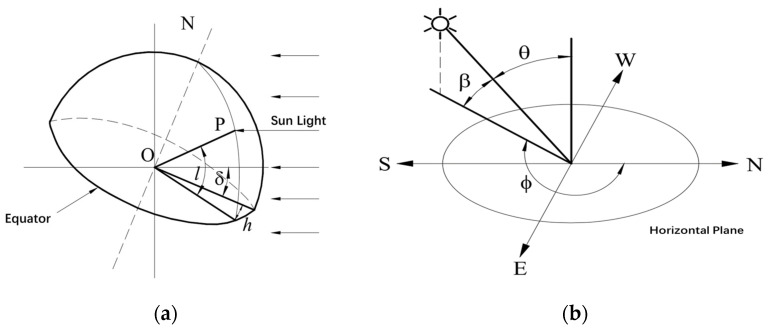
Schematic of sunlight incidence to an object on Earth. (**a**) Plot of geographic latitude l, declination angle δ, and hour angle h; (**b**) sunlight is projected onto a horizontal plane or the ground; (**c**) sunlight projected onto arbitrary surface.

**Figure 4 sensors-24-02341-f004:**
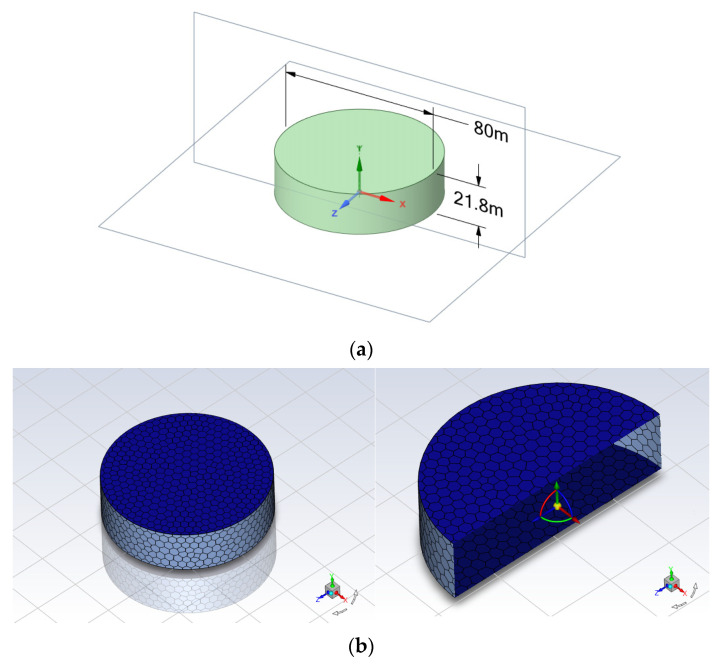
Geometry and meshing numerical model with coordination. (**a**) Geometry model; (**b**) facial meshing in overview and intersection view.

**Figure 5 sensors-24-02341-f005:**
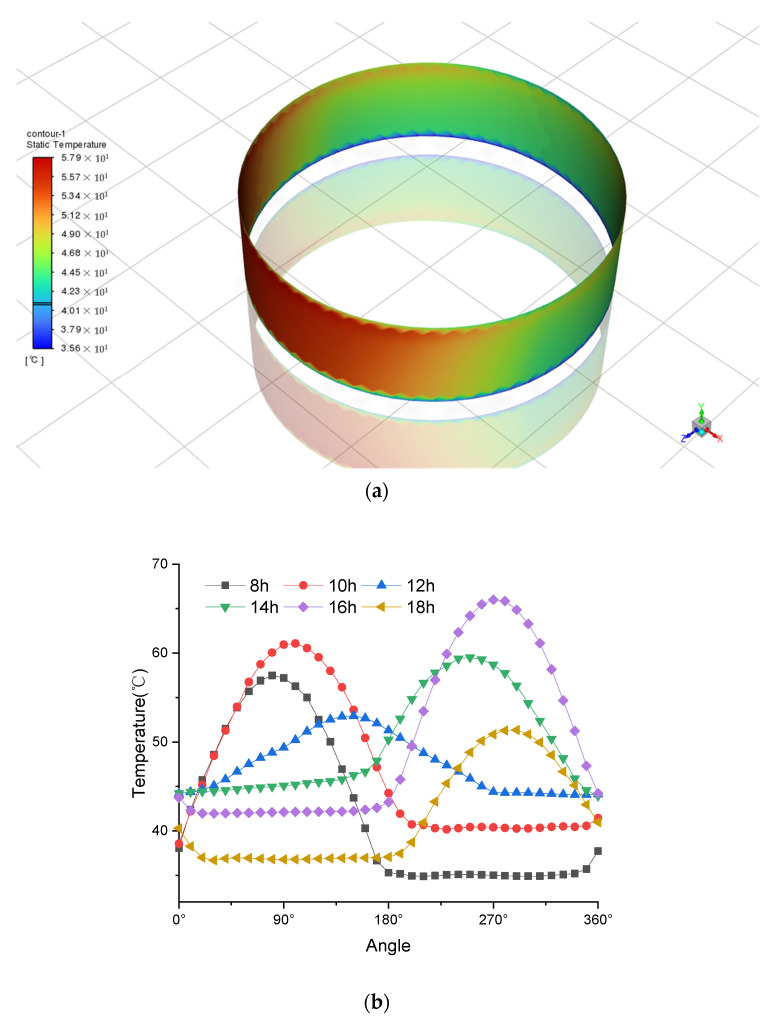
Spatial temperature field underlying solar radiation. (**a**) Temperature cloud map of cylindrical wall at 14:00 o’clock. (**b**) Temperature distribution around the circumference of the oil tank at the height of 1 m where the sensors are installed. (**c**) Data comparison between the simulated temperature and measured temperature at different times at 10, 14, and 16 o’clock.

**Figure 6 sensors-24-02341-f006:**
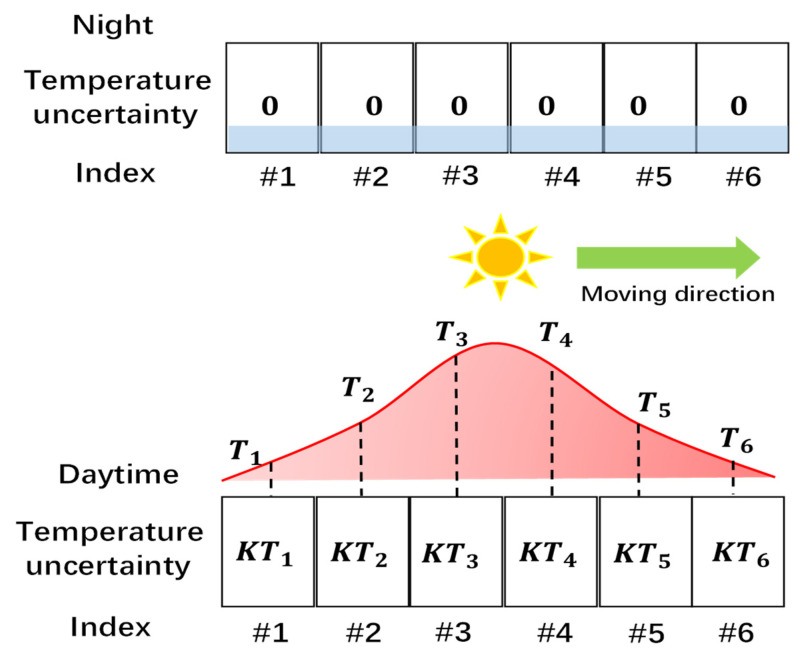
Temperature uncertainty at night and in the daytime. The temperature uncertainty follows the same spatial distribution with temperature.

**Figure 7 sensors-24-02341-f007:**
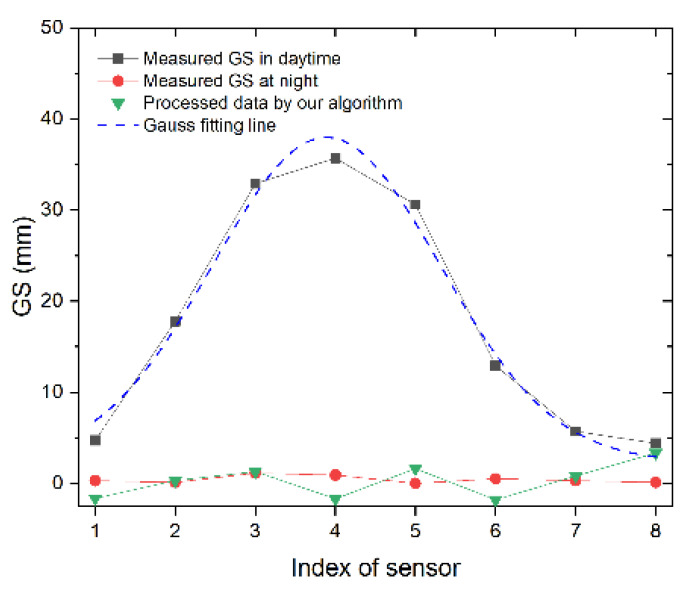
Temperature uncertainty at night and in the daytime.

**Table 1 sensors-24-02341-t001:** Thermal properties of heat transfer model based on steel.

Thermal Property	Value	Unit
Conductivity, k	16.27	$W/\left(m\times K\right)$
Density, ρ	8030	$\mathrm{kg}/m^{3}$
Specific heat, $c_{p}$	502.48	J/kg/℃
Inter-emissivity, ε	0.95	-

**Table 2 sensors-24-02341-t002:** Error analysis before and after the algorithm process.

Index of Sensor	GS in Daytime (mm)	GS at Night (mm)	Processed Data by Algorithm (mm)	Difference between Daytime and Night (mm)	Difference between Processed Data and Night (mm)
#1	4.70	0.30	−1.65	4.40	1.95
#2	17.80	0.10	0.30	17.70	0.20
#3	32.90	1.13	1.29	31.77	0.16
#4	35.70	0.90	−1.69	34.80	2.59
#5	30.60	0.00	1.62	30.60	1.62
#6	12.90	0.50	−1.81	12.40	2.31
#7	5.70	0.30	0.81	5.40	0.51
#8	4.40	0.10	3.33	4.30	3.23

## Data Availability

The data generated or analyzed as part of the research are not publicly available. This research is continuing and the data will be disclosed with the permission of the oil-tank-running corporation.

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
