# Peer review of "Temperature Uncertainty Reduction Algorithm Based on Temperature Distribution Prior for Optical Sensors in Oil Tank Ground Settlement Monitoring"

_sensors, 2024, doi:10.3390/s24072341_

Round 1

Reviewer 1 Report

Comments and Suggestions for Authors

Dear colleagues! Thank you for the opportunity to review the article Temperature uncertainty reduction algorithm based on temperature distribution priority for optical sensors in oil tank ground settlement monitoring of the authors Tao Lau, Tao Jing, Gong Liu and Hansen Sun.

The work is devoted to the study of temperature uncertainties in the ground settlement  monitoring sensor and the influence of temperature, which was evaluated theoretically and based on the results of numerical modeling. The article is good presented and has a clear structure which simplifies the understanding of the research essence.

The article made an exceptionally positive impression on me and I believe that it is already worthy of publication in this form, but I still want to suggest mentioning the following:

1.What is the difference between your research and the previous ones?

2. The fact that you have developed your algorithm will be even more revealing if you indicate which issues were solved better thanks to it than those of the authors who solved this problem before you.

3. What are the perspective of the research ?

4. What are the possibilities of large-scale application of your algorithm and the estimated time frame for implementation?

5.How many times has the experiment been conducted?

6. Is there a dependence on the season?

Best regards

Reviewer 2 Report

Comments and Suggestions for Authors

1. The Abstract did not reflect the content and summarize the problem, method, results, and conclusions.

2. In the introduction section, the logic of thinking in introducing the background and purpose of the study is not clear enough, please give more knowledge or the significance of the study. The following references are suggested to cite:

(1) Shaghaleh, H., Hamoud, Y. A., & Sun, Q. (2024). Functionalized nanocellulose nanocomposite hydrogels for soil and water pollution prevention, remediation, and monitoring: A critical review on fabrication, application properties, and potential mechanisms. Journal of Environmental Chemical Engineering, 111892.

(2) Ta, Q. T. H., Tran, N. M., Tri, N. N., Sreedhar, A., & Noh, J. S. (2021). Highly surface-active Si-doped TiO2/Ti3C2Tx heterostructure for gas sensing and photodegradation of toxic matters. Chemical Engineering Journal, 425, 131437.

(3) Demory, B., Echeveria, L., Tolfa, C., Harrison, S., Khitrov, V., Chang, A. S., & Bond, T. (2024). Real-Time Tracking of Carbon Dioxide Concentration Using an Optical Microsphere Resonator Sensor. Applied Spectroscopy, 00037028241228883.

3. The state-of-the-art work is interesting but sometimes poorly described or explained. It would be especially helpful if the authors could include explanations and opinions on important areas for further research.

4. It is suggested that the author use a table to compare the results of the material in this paper with the previous works.

5. Figure 4, please check this sentence "its volume is 10 000 𝑚3 with a diameter of 80 m and height of 21.8 m" Is the volume should 110 000 𝑚3?

6. Figure 5 c, data at 16 o’clock or 18??. Please provide more details about Figure 6.

7. The sensing mechanism of  optical GS sensor is insufficient.

Round 2

Reviewer 2 Report

Comments and Suggestions for Authors

The authors have revised the manuscript carefully, it can be accepted in its current form.